# Performance of the UNICEF/UN Washington Group tool for identifying functional difficulty in rural Zimbabwean children

Thomas Frederick Dunne[1]*, Jaya Chandna[1,2], Florence Majo[2], Naume Tavengwa[2], Batsirai Mutasa[2], Bernard Chasekwa[2], Robert Ntozini[2], Andrew J. Prendergast[2,3,4], Jean H. Humphrey[2,4], Melissa J. Gladstone[1]

1 Department of Women and Children's Health, Institute of Life Course and Medical Sciences, University of Liverpool, Liverpool, United Kingdom, 2 Zvitambo Institute for Maternal and Child Health Research, Harare, Zimbabwe, 3 Blizard Institute, Queen Mary University of London, London, United Kingdom, 4 Department of International Health, Johns Hopkins Bloomberg School of Public Health, Baltimore, MD, United States of America

* thomas.dunne@nhs.net

**Data Availability Statement:** Data will be freely available as individual participant data with an accompanying data dictionary at http://ClinEpiDB. org. This platform is charged with ensuring that

## Abstract

### Introduction

Over one billion people live with disability worldwide, of whom 80% are in developing countries. Robust childhood disability data are limited, particularly as tools for identifying disability function poorly at young ages.

### Methods

A subgroup of children enrolled in the Sanitation Hygiene Infant Nutrition Efficacy (SHINE) trial (a cluster-randomised, community-based, 2x2 factorial trial in two rural districts in Zimbabwe) had neurodevelopmental assessments at 2 years of age. We evaluated functional difficulty prevalence in HIV-exposed and HIV-unexposed children using the Washington Group Child Functioning Module (WGCFM), comparing absolute difference using chi-squared or Fisher's exact tests. Concurrent validity with the Malawi Developmental Assessment Tool (MDAT) was assessed using logistic regression with cohort MDAT score quartiles, linear regression for unit-increase in raw scores and a Generalised Estimating Equation approach (to adjust for clusters) to compare MDAT scores of those with and without functional difficulty. A 3-step, cluster-adjusted multivariable regression model was then carried out to examine risk factors for functional difficulty.

### Findings

Functional Difficulty prevalence was 4.2% (95%CI: 3.2%, 5.2%) in HIV-unexposed children (n = 1606) versus 6.1% (95%CI: 3.5%, 8.9%) in HIV-exposed children (n = 314) (absolute difference 1.9%, 95%CI: -0.93%, 4.69%; p = 0.14). Functional difficulty score correlated negatively with MDAT: for each unit increase in WGCFM score, children completed 2.6

epidemiological studies are fully anonymized by removing all personal identifiers and obfuscating all dates per participant through application of a random number algorithm to comply with the ethical conduct of human subjects research. Researchers must agree to the policies and comply with the mechanism of ClinEpiDB to access data housed on this platform. In addition, the data are housed on the ClinEpiDB platform at the Zvitambo Institute for Maternal and Child Health Research and available upon request from Ms. Virginia Sauramba (vsauramba@zvitambo.co.zw).

**Funding:** This study was funded through the following grants: Bill and Melinda Gates Foundation (OPP1021542 and OPP1143707; received by JH); UK DFID/UKAID (received by JH); Wellcome Trust, UK (093768/Z/10/Z and 108065/Z/15/Z; received by AP); Swiss Agency for Development and Cooperation (8106727; received by JH); NIH (R01 HD060338/HD/NICHD; received by JH); and UNICEF (PCA-2017-0002; received by JH). The funders had no role in study design, data collection and analysis, decision to publish or preparation of the manuscript.

**Competing interests:** The authors have declared that no competing interests exist.

(95%CI: 2.2, 3.1) fewer MDAT items (p = 0.001). Children from families with food insecurity and poorer housing were more at risk of functional difficulty.

## Interpretation

Functional difficulty was identified in approximately 1-in-20 children in rural Zimbabwe, which is comparable to prevalence in previous studies. WGCFM showed concurrent validity with the MDAT, supporting its use in early childhood.

## Introduction

The World Health Organization (WHO) estimates that over one billion people live with a disability worldwide; at least 93 million are under the age of 15 years [1, 2]. Children with disability have an increased risk of mortality and morbidity [2–4] and decreased school attendance [2, 5, 6]. Globally, it is estimated that 80% of individuals, and 95% of children under-five, with a disability live in low- and middle-income countries (LMIC) [7]. Recent advances in medical care have resulted in improved child survival and an increasing number of children living with disability [8, 9].

Defining the prevalence of disability in young children is important to advocate for better support and services [10–12]. The United Nations Convention for the Rights of the Child (UNCRC) (1989) sets out the importance of equal inclusion and a right to education for all children, including those with disabilities [13]. This is hindered by poor early identification of children with disabilities and by limited accurate data on prevalence, particularly in the early years [9, 14].

One major issue is that data on prevalence of childhood disability is often incomparable between studies due to differing tools and methodologies [15]. Previously, the Ten Question (TQ) questionnaire was commonly used to screen for childhood disability [16]; however, this tool has been criticised due to its dichotomous response options, low specificity for certain disabilities and lack of validation across all childhood ages. To address this, the UN Washington Group and UNICEF produced The Washington Group/UNICEF Module on Child Functioning (WGCFM) [17]. The WGCFM is a parent-reported survey module designed to identify children aged 2–17 years with functional difficulty across several domains, using multiple response options. It was designed to improve upon the TQ screening tool and Washington Group Short Set [18], for use in population surveys [17, 19]. The term 'functional difficulty' is used to reflect the fact the tool is not diagnostic, but aims to identify individuals 'at risk of exclusion in unaccommodating environments' [17]. It assesses functional difficulty in a number of domains based on the International Classification of Functioning, Disability and Health (ICF).

To date, there are relatively few published studies on use of the WGCFM, although it is now being utilised regularly by many countries in the Multiple Indicator Cluster Surveys (MICS) [15, 20–23]. Prior to its finalisation in 2016, the tool underwent cognitive testing in six countries, followed by field testing of the final draft in Samoa (2014), Mexico (2015) and Serbia (2016) [21, 24, 25]. From this field testing, a cut-off score for functional difficulty in children was created. Furthermore, researchers from South Africa [20], compared scores on WGCFM to the Ages and Stages Questionnaire Third Edition (ASQ-III) and demonstrated comparable results between the tools in a sample of 50 children registered as receiving a grant for Care Dependency, Child Support or Foster Care. These researchers concluded that the WGCFM

could be considered as a tool for surveying childhood disability in larger scale studies. Much larger studies from Cameroon and India (1713 and 1101 children respectively, aged from 2–17 years) found comparable results of functional difficulty in the two countries, and a study in Fijian schools reported good diagnostic accuracy [15, 22, 23]. Studies have highlighted a high rate of 'some difficulty' being reported across multiple domains, questioning the specificity of this response.

We set out to estimate the prevalence of total and severe disability using the WGCFM score among a cohort of HIV-exposed and HIV-unexposed children evaluated in the Sanitation Hygiene Infant Nutrition Efficacy (SHINE) trial in rural Zimbabwe. We aimed, first, to evaluate the validity of the WGCFM for a cohort of HIV-unexposed, young children who were concurrently assessed using the Malawi Developmental Assessment Tool (MDAT), a directly observed tool for measuring child development, which is well-validated for use in Africa. Second, we used data from a cohort of children born to HIV-positive mothers, to compare disability prevalence between children exposed and unexposed to HIV. Finally, we aimed to explore factors associated with risk of disability as defined by the WGCFM.

## Methods

### SHINE trial

The SHINE trial was a cluster-randomised, community-based, 2x2 factorial trial in two rural districts in Zimbabwe. The primary aim was to assess the effects of improved water, sanitation and hygiene (WASH), and/or improved infant and young child feeding (IYCF) on child linear growth and haemoglobin at 18 months of age [26]. The full SHINE protocol and analysis plan are available at https://osf.io/w93hy/.

Trial recruitment took place between November 2012 and March 2015. Village Health Workers identified pregnancies through prospective surveillance. Pregnancy tests were undertaken for any women who reported missing a menstrual period. If positive, a research nurse visited the home and repeated the urinary pregnancy test and enquired about any recent vaginal bleeding. Any women with vaginal bleeding in the last 2 weeks were not enrolled; they were referred to clinics and re-visited 2 weeks later to ascertain whether they were still pregnant. Inclusion criteria were confirmed pregnancy, permanent residency in one of the trial clusters and the provision of written, informed consent. Child outcomes were measured at 1, 3, 6, 12 and 18 months of age; a sub-study at 24 months of age evaluated early child development (ECD) and disability. Details of all outcomes measured can be found in the original SHINE trial publication [26].

### Data collection tools

ECD was assessed at 24 months of age (allowable range 102–112 weeks of age) using a panel of tests as previously described [27]. All tools were translated into Shona and Ndebele and back-translated into English. The current study reports functional difficulty as defined by the WGCFM, and ECD as defined by MDAT.

### The Washington Group/UNICEF Module on Child Functioning (WGCFM)

The WGCFM is a parent-report survey module designed to identify children aged 2–17 years with functional difficulty across multiple domains [17]. There are two versions of the tool, for use in 2-4-year-olds or 5-17-year-olds, with different domains assessed depending on the age of the child. This study utilised the 2015 version of the tool for 2-4-year-olds, comprising ten questions assessing vision, hearing, mobility, communication, learning, playing and

controlling behaviour. Questions include: 'Compared with children of the same age, does (name) have difficulty walking?', 'Does (name) have difficulty understanding you', 'Compared with children of the same age, does (name) have difficulty playing with toys or household objects?'. Each question is scored on a Likert scale from 1–4, equating to 'no difficulty', 'some difficulty', a lot of difficulty', and 'cannot do at all', respectively (5 = don't know). The score relates to a binary classification of functional difficulty or no functional difficulty. The final cut-off defined by the Washington Group in field testing was a score of 3 (indicating 'a lot of difficulty'), in any domain except for 'controlling behaviour'. This domain detected a significantly higher functional difficulty prevalence than the others, therefore a higher cut-off was used to reduce the risk of false-positives [21]. In our trial, inclusion of the controlling behaviour domain detected a high prevalence of functional difficulty. For this reason, it was excluded and the remainder of the WGCFM was used to define functional difficulty in this population.

In this study, the definition of functional difficulty was therefore a response of 'a lot of difficulty' in any of any of the 9 questions of the tool (excluding the controlling behaviour domain). The definition of severe functional difficulty was a response of 'cannot do at all' for any of the 9 questions.

## The Malawi Developmental Assessment Tool (MDAT)

The MDAT is a culturally appropriate tool for assessing child development in rural African settings [28]. It assesses four domains: 1) gross motor coordination (36 items), 2) fine motor coordination (36 items), 3) social (30 items), and 4) language (36 items), with 138 items in the published version of the tool. The latter three domains measure components of cognitive development and the tool shows good sensitivity, reliability and validity in detecting neurodisabilities [28]. It produces a continuous score for development (with a higher score indicating a higher level of development). Prior to the SHINE trial, the MDAT underwent translation, back-translation and piloting for use in rural Zimbabwe. This led to some minor changes to the wording of questions and items included in the kit, e.g. the items of tools for children to name, to ensure cultural appropriateness.

## Data collection methods

Data collection was conducted by a team of 11 research nurses. The ECD assessment took 2–3 hours and was conducted in the child's home. The time of the visit was chosen to suit the household, and breaks of any duration were allowed where the family had other commitments or children needed rest. Data were collected on paper forms and manually inputted into tablets by the research nurses. Children found to have disability were referred to local clinics for assessment and further management. Several approaches to validation and quality control were undertaken, as previously described (S1 Methods) [27].

## Statistical analysis

Functional difficulty prevalence was calculated using the WGCFM and further explored through examining each individual domain. Functional difficulty prevalence was compared between HIV-exposed and HIV-unexposed children using chi-square or Fisher's exact tests, and absolute difference reported.

We adopted the same approach as UNICEF in handling missing data (Mitch Loeb, personal communication). Where a patient had any missing data, they were excluded from analysis; we extended this to also exclude those who selected 'don't know' as a response. A sensitivity analysis was performed in which children who were excluded due to missing values (children with fewer than 10 but at least one response) were included in analysis to explore whether this

differing approach resulted in a significant difference in functional difficulty prevalence. In this analysis, a missing domain value was assumed to be no functional difficulty in that domain.

To assess concurrent validity of the WGCFM, scores were compared to MDAT scores. Firstly, the cohort was divided by MDAT score quartiles, with quartile 1 being 'least developed' and quartile 4 being 'most developed'. The prevalence of functional difficulty and domain-specific difficulty was then assessed by logistic regression, comparing the prevalence of functional difficulty in the second, third and fourth quartiles, separately, to the first quartile. Secondly, linear regression was used to assess for change in MDAT score per unit increase in raw WGCFM score. Thirdly, MDAT scores were compared between those with and without functional difficulty and severe functional difficulty, as defined by WGCFM. This utilised a Generalised Estimating Equation approach to adjust for clusters (subject variable: cluster, working correlation matrix: exchangeable, distribution: Gaussian).

Risk factors for functional difficulty used a previously published, 3-step approach [29].

- Step 1: univariable associations between each factor and functional difficulty were assessed using an appropriate statistical test (e.g. t-test, $chi^2$, Kruskal-Wallis). For robustness of comparison estimate, univariable associations were also explored using a Generalised Estimating Equation to adjust for cluster effect. Variables were included in step 2 of the model if they met the significance cut-off of $p < 0.1$.

- Step 2: variables with a univariable association of significance of at least $p < 0.1$ were entered into a multivariable model.

- Step 3: variables that remained significant at $p < 0.1$ in step 2 were entered in to a final multivariable model along with a priori variables: age, trial arm, season, data collector and birthweight, due to a recognised independent association between very low birthweight and disability [30, 31]. Those remaining significant in the final model ($p < 0.05$) were identified as risk factors for functional difficulty. *A* priori variables were included to reduce bias; although adjusted for in step 3, they are not reported in the final model. For robustness of estimates, each step of the regression was cluster-adjusted using a Generalised Estimating Equation approach with an exchangeable correlation structure. In addition to this, the regression was run a second time with a univariable association inclusion cut off of $p < 0.2$, to ensure the model was not inappropriately filtering out important variables at this early stage; this approach did not alter the variables included in the final model output.

Analyses were undertaken using SPSS [32] and STATA [33].

### Ethics and trial registration

The trial protocol was approved by the Medical Research Council of Zimbabwe and the Institutional Review Board of the Johns Hopkins Bloomberg School of Public Health. The trial statistical analysis plan included outcomes of ECD (see https://osf.io/w93hy). The SHINE trial was registered at Clinical-Trials.gov (NCT01824940). Written informed consent for participation was obtained from the primary caregiver (parent or guardian) of all children who took part in this substudy.

## Results

### Recruitment

5280 pregnant women were enrolled to the SHINE trial at median 12 (IQR 9, 16) gestational weeks between November 22nd 2012 –March 27th 2015. Fig 1 depicts the study CONSORT

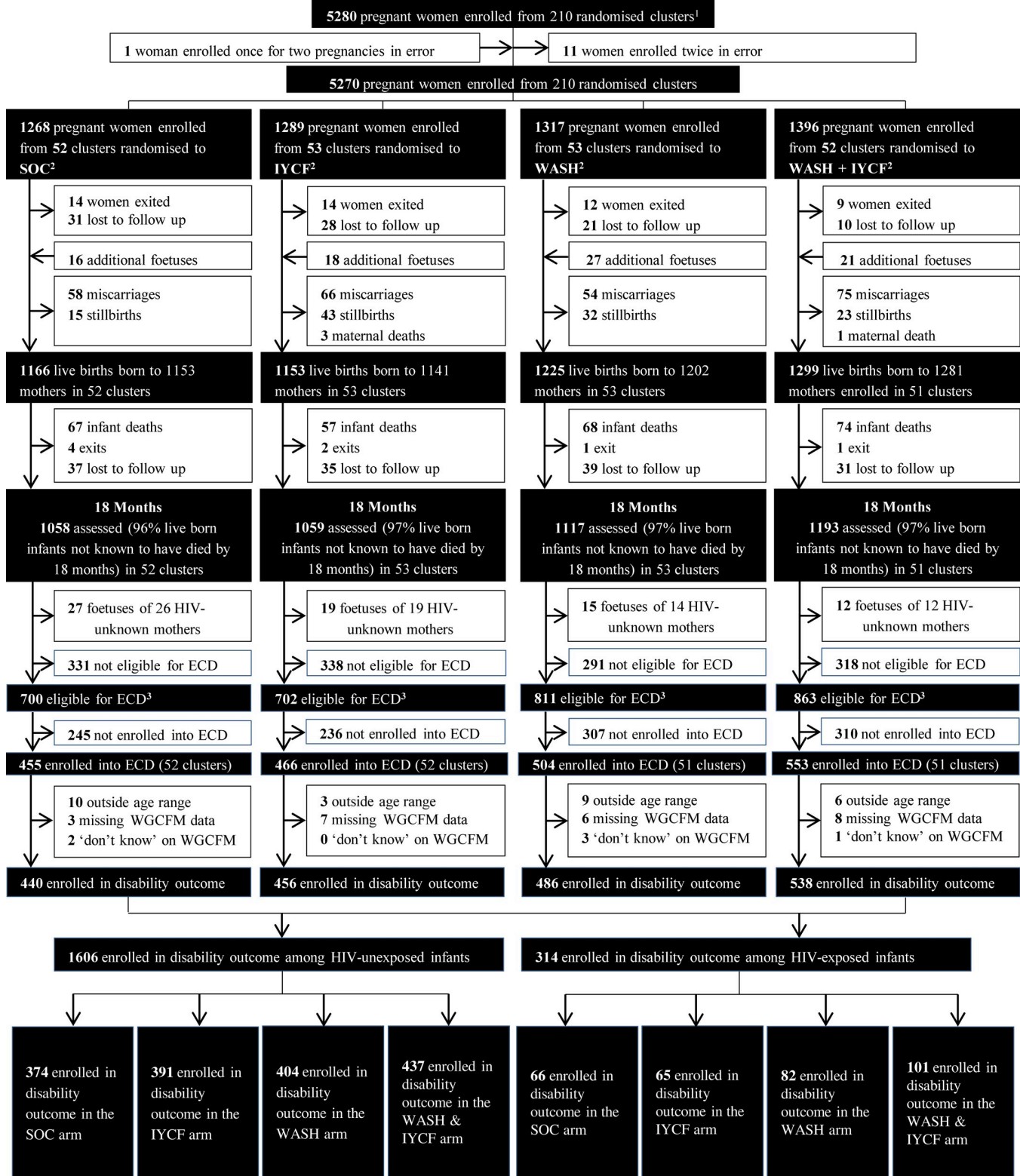

**Fig 1. CONSORT flow diagram.** 212 clusters were randomised, 53 in each of the four trial arms. After randomisation, one cluster was excluded as it was determined to be in an urban area, one cluster was excluded as the VHW covering it mainly had clients outside the study area, and one more was merged into a neighbouring cluster based on subsequent data on VHW coverage. Three new cluster designations were created due to anomalies in the original mapping; for two of these, the trial arm was clear—the third contained areas that were in two trial arms, and was assigned to the underrepresented arm, resulting in 53

clusters in each arm. All of this occurred before enrolment began. When enrolment was completed, however, there were two clusters (SOC, and WASH+IYCF) in which no women were enrolled, leaving a total of 210 clusters available for analysis. [2] SOC = Standard of Care; IYCF = Infant and Young Child Feeding; WASH = Water and Sanitation/Hygiene. [3] In SOC arm, includes 3 infants who were not eligible but enrolled into ECD; In IYCF arm, includes 0 infants who were not eligible but enrolled into ECD; In WASH arm, includes 1 infant who were not eligible but enrolled into ECD; In WASH+ IYCF arm, includes 4 infants who were not eligible but enrolled into ECD.

flow diagram and shows the study flow between recruitment and disability assessments at age 24 months. 1978 infants from this cohort were enrolled into the ECD substudy at 24 months; of these, 28 were outside the age window, 24 did not answer all questions of the WGCFM and 6 selected 'don't know' on one or more options of the WGCFM. There were no missing MDAT results for any participants. This left a cohort of 1920 children, of whom 1606 were HIV-unexposed and 314 HIV-exposed.

## Baseline characteristics of sample included in WGCFM assessment

Table 1 reports baseline characteristics for children enrolled in the disability analysis (n = 1920). S1 Table reports baseline differences between those who had ECD assessments compared to those who did not.

## Functional difficulty assessed by WGCFM

Functional difficulty prevalence among HIV-unexposed children was 4.2% (95% CI: 3.2, 5.2%) (n = 67). Of those with a functional difficulty, 16.4% (95%CI: 9.0, 25.4%) (n = 11) had a severe functional difficulty (cohort prevalence 0.7% (95% CI: 0.3, 1.1%); (Table 2). S2 Table shows a breakdown of WGCFM responses by question. Following the excluded 'controlling behaviour' domain, most functional difficulty was driven by the 'playing' domain, followed by 'learning and cognition'. S1 Fig compares the prevalence of functional difficulty by each cut-off during field testing of the tool in Mexico, Samoa and Serbia, compared with findings from the current study [21].

## Maternal HIV status and disability

Complete data were collected from 314 children born to HIV-positive mothers. Functional difficulty prevalence was 6.1% (95%CI: 3.5%, 8.9%) (n = 19) and severe functional difficulty prevalence 0.3% (95%CI: 0.0%, 0.9%) (n = 1). There was no significant difference in the prevalence of functional difficulty (absolute difference 1.9% (95%CI: -0.93%, 4.69%), p = 0.14) or severe functional difficulty (absolute difference -0.4% (95%CI: -1.1%, 0.38%), p = 0.45) between the HIV-exposed and HIV-unexposed cohorts.

## Missing data sensitivity analysis

Nineteen participants were excluded from analysis for missing data, with one child having no recorded values for the WGCFM. A further five participants who had selected 'don't know' for at least one option, were excluded. This totalled an additional 24 respondents with at least one domain of non-missing. Inclusion of these participants gave a functional difficulty prevalence of 4.3% (95%CI: 3.3%, 5.3%) (n = 70).

## WGCFM and MDAT concurrent validity

Table 3 shows the prevalence of functional difficulty in the SHINE cohort by MDAT score quartiles. For all domains except 'playing', functional difficulty prevalence was highest in those with the lowest MDAT score, with no participants in the top 50% of MDAT scores having a

**Table 1. Baseline characteristics of mothers and children in the disability sub study of the Sanitation Hygiene Infant Nutrition Efficacy (SHINE) trial.**

| Baseline Characteristic | Enrolled into disability study |
|---|---|
| Women assessed, N | 1902 |
| Children assessed, N | 1920 |
| Women completing baseline visit, N | 1814 |
| **Household characteristics** | |
| Size, median (IQR) [n] | 5 (3,6) [1801] |
| Wealth quintile, percent [n] | |
| Lowest | 18.9 [341] |
| Second | 19.4 [350] |
| Middle | 21.0 [379] |
| Fourth | 21.2 [382] |
| Highest | 19.5 [351] |
| *Sanitation* | |
| Any latrine at household, percent [n] | 36.8 [648] |
| *Water* | |
| Main source of household drinking water is improved, percent [n] | 62.4 [1102] |
| *Hygiene* | |
| Handwashing station at household, percent [n] | 9.7 [163] |
| Improved floor, percent [n] | 53.7 [956] |
| *Diet quality and food security* | |
| Household meets minimum dietary diversity score, percent [n] | 40.9 [644] |
| Coping Strategies Index, median (IQR) [n] | 1 (0,8) [1763] |
| **Maternal characteristics** | |
| Age, years, mean (SD) [n] | 27.2 (6.8) [1705] |
| Height, cm, mean (SD) [n] | 159.8 (9.5) [1863] |
| MUAC, cm, mean (SD) [n] | 26.5 (3.1) [1881] |
| Completed schooling, years, mean (SD) [n] | 9.5 (1.8) [1788] |
| Parity, median (IQR) [n] | 2 (1,3) [1406] |
| Married, percent [n] | 95.6 [1699] |
| Employed, percent [n] | 9.5 [170] |
| Religion, percent [n] | |
| Apostolic | 49.8 [890] |
| Other Christian (Pentecostal, Catholic, other Christian) | 43.3 [774] |
| Other religions (Muslim and other) | 6.9 [124] |
| Maternal capabilities | |
| Gender norms and attitudes, mean (SD) [n] | 2.3 (0.8) [1792] |
| Perceived social support, mean (SD) [n] | 3.6 (0.6) [1766] |
| Perceived physical health, mean (SD) [n] | 3.4 (1.0) [1574] |
| Mothering self-efficacy, mean (SD) [n] | 4.0 (0.4) [1774] |
| Perceived time stress, mean (SD) [n] | 2.7 (0.7) [1767] |
| Decision-making autonomy, median (IQR) [n] | 5 (4.5) [1609] |
| **HIV status, percent [n]** | |
| Positive | 16.4 [311] |
| Negative | 83.7 [1591] |
| Unknown | 0.0 [0] |
| **Infant characteristics** | |
| Female, percent [n] | 50.0 [959] |

(*Continued*)

**Table 1.** (Continued)

| Baseline Characteristic | Enrolled into disability study |
|---|---|
| Birth weight, kg, mean (SD) [n] | 3.1 (0.5) [1830] |
| Birth weight <2500g, percent [n] | 8.9 [162] |
| Institutional delivery, percent [n] | 89.3 [1624] |
| Vaginal delivery, percent [n] | 92.7 [1729] |

functional difficulty based on these questions. By contrast, the playing domain showed a positive relationship between WGCFM and MDAT, with those in the highest quartile by MDAT score being over twice as likely to have a functional difficulty than those with the lowest MDAT score.

Logistic regression was used to further explore the change in prevalence of functional difficulty across MDAT quartiles. Table 4 reports unadjusted odds ratios for overall functional difficulty and the specific 'walking', 'learning and cognition' and 'playing' domains of the WGCFM. There were no children who had a functional difficulty identified in the 'seeing' and 'hearing' domains, and the numbers of children scoring positive in the 'relationship' and 'communication' domains were too small to undertake regression analyses. Participants in the second compared to the first MDAT quartile were less likely to have a functional difficulty in the 'walking' (OR 0.13: 95%CI 0.02, 1.02) or 'learning and cognition' (OR 0.21: 95%CI 0.05, 0.95) domains. In the 'play' domain this relationship was reversed: children with the highest MDAT scores for ECD were significantly more likely to have a functional difficulty than those in the lowest MDAT quartile (OR 2.84: 95%CI 1.28, 6.31). For overall functional difficulty, those in the third quartile of MDAT score were significantly less likely to have a functional difficulty than those in first quartile (OR 0.43: 95%CI 0.20, 0.91). This relationship was no longer apparent when comparing fourth quartile to first quartile (OR 0.98: 95%CI 0.54, 1.80).

The cohort mean MDAT score was 92.7 (SD 9.4). Each unit increase in raw WGCFM score was associated with a significant reduction in MDAT score ($\beta$ = -2.65 (95%CI -3.11, -2.20), p<0.001). This means children completed 2.7 fewer items on the MDAT (from a total of 138 items) for every unit rise of WGCFM. This remained true across each domain: gross motor -0.51 (95%CI -0.67, -0.35), fine motor -0.64 (95%CI -0.76, -0.52), social -0.58 (95%CI -0.70, -0.46), and language -0.93 (95%CI -1.14, -0.72); all p<0.001. Participants with a functional difficulty had a significantly lower MDAT score than those without, B (cluster adjusted mean difference) = 4.40 (SE 2.14), p = 0.04. This translates to children with a functional difficulty having a mean MDAT score of 4.4 points lower than those without. This difference became more significant when comparing those with severe functional difficulty to no severe functional difficulty (B = 29.54 (SE 5.79), p<0.001) and severe functional difficulty to no functional difficulty (B = 29.97 (SE 6.13), p<0.001).

## Risk factors for functional difficulty

Table 5 reports risk factors for functional difficulty in this cohort. Two factors were identified as independent predictors of the risk of functional difficulty: housing quality as denoted by floor type, and food security coping strategies index (CSI). Having an improved floor (as opposed to an unimproved/mixed floor type) was associated with a decreased risk of functional difficulty (B = -0.795 (95%CI -1.51, -0.08), p = 0.029). CSI is a measure of how often strategies are used to cope with food insecurity, and how severe these strategies are; a higher value indicates a higher degree of food insecurity. A higher CSI score was associated with a higher risk of functional difficulty (B = 0.017 (95%CI 0.002, 0.03), p = 0.023).

**Table 2. Table reporting moderate, severe and total functional difficulty prevalence in HIV unexposed infants.**

| WGCFM Functional Difficulty Level | Frequency | % of total participants (n = 1606) | % of those with functional difficulty (n = 67) |
|---|---|---|---|
| **Moderate** | 56 | 3.5 | 83.6 |
| Male | 31 | 1.9 | 46.3 |
| Female | 25 | 1.6 | 37.3 |
| **Severe** | 11 | 0.7 | 16.4 |
| Male | 7 | 0.4 | 10.5 |
| Female | 4 | 0.3 | 6.0 |
| **Total with disability** | 67 | 4.2 | 100 |

Factors that were entered into the model at step 1 were household size, socio-economic status score, socio-economic status quintile, latrine, improved water, improved floor, dietary diversity score, minimum dietary diversity score, diet quality and food security coping strategies index, maternal age, maternal height, maternal mean upper arm circumference, maternal education, maternal parity, maternal marital status, maternal employment, maternal religion, Edinburgh Postnatal Depression Scale raw score, maternal depression, maternal capabilities (gender norms and attitudes, perceived social support, perceived physical health, mothering self-efficacy, perceived time stress, decision-making autonomy), infant sex, gestation, delivery location, mode of delivery, length-for-age at 24 months (z score), weight-for-age at 24 months (z score), head circumference-for-age at 24 months (z score) and weight-for-length at 24 months (z score). Introduced at step 3 were birth weight, trial arm, season, data collector and absolute age at time of test completion. Factors reported in the table above are those found to be significant in the output of step 3 along with the intercept.

## Discussion

This study reports the prevalence of functional difficulty in a cohort of rural Zimbabwean children at 2 years of age. Disability was defined by the WGCFM, with concurrent validity assessed using the MDAT as a comparator. This aimed to address important gaps in the literature on the performance of the WGCFM, particularly at young ages. Following removal of the 'controlling behaviour' domain, this study found that 4.2% (95%CI 3.2%, 5.2%) of HIV-unexposed children were categorised as having a functional difficulty, a figure that is consistent with general functional difficulty prevalence from other studies [21]. A stricter cut-off with the WGCFM tool was used to examine 'severe functional difficulty', with findings reflecting prevalence reported in previous studies. Functional difficulty prevalence was 1.9% higher in children with HIV exposure (6.1% (95%CI 3.5%, 8.9%)), however this difference was not statistically significant. WGCFM for 2-4-year-olds showed concurrent validity with the MDAT for

**Table 3. Table reporting functional difficulty prevalence by Washington Group Child Functioning Module domain through increasing quartiles of Malawi Developmental Assessment tool (MDAT) score.** HIV unexposed infants included only, n = 1606.

| Quartile (MDAT score) [n] | Functional Difficulty by Domain % (n) | | | | | | | | Overall Functional Difficulty excluding Q10 | Overall Severe Functional Difficulty excluding Q10 |
|---|---|---|---|---|---|---|---|---|---|---|
| | Seeing: Q1 | Hearing: Q2 | Walking: 3Q | Learning and Cognition: Q4, Q8 | Relationships: Q5 | Communication: Q6,7 | Playing: Q9 | Controlling Behaviour: Q10 | | |
| **1** (87) [451] | 0.0 (0) | 0.0 (0) | 2.0 (9) | 2.4 (11) | 0.9 (4) | 1.3 (6) | 2.0 (9) | 14.9 (67) | 5.5 (25) | 2.2 (10) |
| **2** (92) [383] | 0.0 (0) | 0.0 (0) | 0.3 (1) | 0.5 (2) | 0.0 (0) | 0.0 (0) | 2.4 (9) | 10.2 (39) | 3.1 (12) | 0.3 (1) |
| **3** (99) [406] | 0.0 (0) | 0.0 (0) | 0.0 (0) | 0.0 (0) | 0.0 (0) | 0.0 (0) | 2.5 (10) | 14.0 (57) | 2.5 (10) | 0.0 (0) |
| **4** (125) [366] | 0.0 (0) | 0.0 (0) | 0.0 (0) | 0.0 (0) | 0.0 (0) | 0.0 (0) | 5.5 (20) | 13.7 (50) | 5.5 (20) | 0.0 (0) |

**Table 4. Table reporting odds of functional difficulty across quartiles of increasing MDAT score.**

| | | Likelihood of Functional Difficulty by MDAT Quartile | | | | | | | |
|---|---|---|---|---|---|---|---|---|---|
| | | Functional Difficulty (Q1-Q9) | | Walking* | | Learning and Cognition* | | Playing | |
| | | Odds ratio (95% CI) | P value | Odds ratio (95% CI) | P value | Odds ratio (95% CI) | P value | Odds ratio (95% CI) | P value |
| MDAT Quartile | 1 | 1.0 | | 1.0 | | 1.0 | | 1.0 | |
| | 2 | 0.55 (0.27, 1.11) | 0.096 | 0.13 (0.02, 1.02) | 0.052 | 0.21 (0.05, 0.95) | 0.043 | 1.18 (0.46, 3.01) | 0.726 |
| | 3 | 0.43 (0.20, 0.91) | 0.027 | | | | | 1.24 (0.50, 3.08) | 0.643 |
| | 4 | 0.98 (0.54, 1.80) | 0.961 | | | | | 2.84 (1.28, 6.31) | 0.011 |

* no children were identified as having a functional difficulty based on 'walking' or 'learning and cognition' in MDAT quartiles 3 and 4 therefore no odds could be calculated.

identifying functional difficulty. An inverse relationship between the tools was identified, with higher functional difficulty on WGCFM being associated with lower MDAT scores (and a completion of 2.6 fewer items on the MDAT per unit increase in raw functional difficulty score). Regression analysis identified poor food security and poor housing to be associated with risk of functional difficulty in this population. Collectively, these findings support use of the WGCFM tool in 2-year-old children in rural Zimbabwe.

The prevalence of functional difficulty in this cohort (4.2% (95%CI 3.2%, 5.2%)) was similar to that reported in Samoa (2.8%), Serbia (3.8%) and Mexico (5.4%) [21]. Values of functional difficulty remained comparable when exploring the range of cut-offs for defining functional difficulty with the WGCFM that were trialled during field testing (S1 Fig). These findings suggest some cross-cultural validity of the WGCFM to provide a rapid assessment of functional difficulty for use in research, surveys and censuses. Researchers should be wary of how data should be used at an individual level, as the tool is designed to provide a binary outcome of 'functional difficulty' or 'no functional difficulty', giving little information on the specific area of disability. For example, a raw score of 4 could indicate that a child has 'some difficulty' across four separate domains, or that they are completely blind. Our findings corroborate the concerns of previous studies around the false-positive rate when using the 'controlling behaviour' domain question for functional difficulty in this age group [15, 21]; this concern initially prompted a more stringent cut off for the controlling behaviour domain in the tool for 2-4-year-olds [21].

Using a stricter cut-off, we defined a 'severe functional difficulty' prevalence of 0.7%, in keeping with studies from Serbia, Mexico and Samoa (0.0%– 0.8%). Severe functional difficulty in this SHINE substudy was only reported with children who scored in the first two MDAT quartiles (the individuals with the lowest MDAT scores for ECD). MDAT scores showed a statistically significant difference when comparing those with and without severe functional difficulty, demonstrating strong concurrent validity with MDAT. The strength of the relationship between severe functional difficulty and MDAT score supports the inclusion of a second, more stringent, cut-off to identify children requiring greater support.

**Table 5. Table reporting final parameters included in multivariable regression analysis for risk of functional difficulty.**

| Parameter | B | Std. Error | 95% Confidence Interval | | Hypothesis Test | | 95% Confidence Interval for Exp(B) | | |
|---|---|---|---|---|---|---|---|---|---|
| | | | Lower | Upper | Chi$^2$ | Significance | Exp(B) | Lower | Upper |
| Intercept | 8.886 | 10.8433 | -12.366 | 30.139 | 0.672 | 0.412 | 7232.797 | 4.261E-6 | 1.228E13 |
| Improved floor | -0.795 | 0.3645 | -1.510 | -0.081 | 4.761 | 0.029 | 0.451 | 0.221 | 0.922 |
| Coping Strategies Index | 0.017 | 0.0077 | 0.002 | 0.033 | 5.146 | 0.023 | 1.018 | 1.002 | 1.033 |

Functional difficulty was assessed in 314 children born to HIV-positive mothers. We previously reported lower ECD scores in HIV-exposed compared to HIV-unexposed children in this cohort [34]. We hypothesised that there would be a higher risk of disability among HIV-exposed children. Functional difficulty prevalence was 1.9% higher in the HIV-exposed group compared to the HIV-unexposed (6.1% vs 4.2%); this difference was however not statistically significant. This study was underpowered to assess HIV exposure as a risk factor. This relationship requires further investigation and validation in larger studies.

Overall, the inverse relationship between MDAT and WGCFM score demonstrates an example of good concurrent validity for identifying functional difficulty. Conversely, functional difficulty prevalence by the 'controlling behaviour' domain in our cohort showed no logical relationship with MDAT quartiles (prevalence in quartiles one to four were 14.86%, 10.18%, 14.04% and 13.66% respectively). Some possible reasons for false-positives in this domain could be: reporting of normal tantrums in young children as 'controlling behaviour' or lack of understanding of what 'controlling behaviour' means. Despite our efforts in translation and back translation, more cognitive testing on the understandability of this question in African settings could be conducted, particularly for this young age group. This study also detected a high prevalence of functional difficulty in the 'playing' domain, assessed by the question 'Compared with children of the same age, does (name) have difficulty playing with toys or household objects?'. As with the 'controlling behaviour' domain, functional difficulty in playing did not correlate with the MDAT. We found a small increase in functional difficulty prevalence through the first three MDAT quartiles followed by a sharp increase, whereby those in the highest quartile for MDAT score were more than twice as likely to have a functional difficulty in the play domain than those in the first quartile. It is possible that this item was confusing for parents to answer in this setting, with a lack of clarity on how to respond if a child had nothing to play with, or played with items not included in 'toys or household objects'. Furthermore, in some settings, parents may not utilise the term 'play' in the same way, and a differing view of the importance of play in relation to child development may influence the response provided by a caregiver [35, 36]. This question has been simplified in the final version of the tool, now reading: 'Compared with children of the same age, does (name) have difficulty playing?'. It is possible that this simplification may overcome some of this confusion. Further testing of the final version of the WGCFM will hopefully provide further insight into validity of the play domain.

Multivariable regression showed an increased risk of functional difficulty in children with less food security and poorer housing (flooring). Families with more money are more likely to have better quality flooring in their homes and greater food security. It is likely that families with a more stable financial situation have better access to protective factors such as healthcare, education and specialist toys/learning materials.

Our study had strength and weaknesses. The SHINE study provided a large cohort of children with clear HIV exposure categorisation. Furthermore, there was a tight window of ages at assessment, and a range of tools were used, providing a rich description of demographic factors. Our study was limited by use of a draft version of the WGCFM (due to availability when commencing the sub-study in 2014) which differs slightly from the final version of the tool now in use, in the number of screening questions and some differences in wording. Despite translation and back-translation of the tools, there remained a risk that cultural differences may have impacted interpretation of the WGCFM and, in turn, impacted our results. Furthermore, as the same assessors completed ECD assessments as did the WGCFM, this may have risked the introduction of response bias. Finally, we did not conduct a gold standard, clinical assessment for disability in children, despite our use of a comparison tool with known cultural appropriateness for assessing ECD in rural Africa. The use of the term 'disability' itself poses potential issues due the range of possible interpretations stemming from its biological,

psychological and social associations. The WHO state 'Disability refers to the interaction between individuals with a health condition (e.g., cerebral palsy, Down syndrome and depression) and personal and environmental factors (e.g., negative attitudes, inaccessible transportation and public buildings, and limited social supports)' [1]. This definition highlights the importance of understanding and labelling disability within the wider context. Clearly, the WGCFM and MDAT mainly assess the functional components of disability but may not necessarily grapple with the full picture. We must be aware of this when considering and discussing the use of a tool for measuring disability. It is therefore vital, that we advocate for tools that also include other elements which may represent disability along the spectrum of the International Classification of Functioning and Disability in Children and Youth (ICF-CY). This would include measures of participation and technological and environmental support [37].

In summary, the results of this study indicate that the WGCFM shows concurrent validity with the MDAT for identifying functional difficulty in 2-year-old children in rural Zimbabwe. WGCFM is simple to use, and these findings support its use in providing a rapid assessment of functional difficulty for use in research, surveys and censuses. Further work should look into the use of the 'controlling behaviour' domain in such a young age group and consider modification or removal of the question. This study also suggests exploration of the 'playing' domain in children of this age in a rural environment in Africa. Future studies utilising the WGCFM could explore if a) the tool continues to be valid for this age group in other populations, b) if there is a practical use of having a third WGCFM category of severe functional difficulty. This could identify those at severe risk of exclusion from integration into society due to functional difficulty, highlighting them as individuals requiring immediate further investigation and/or support. Future studies should continue to assess population-specific risk factors for functional difficulty, including HIV exposure; this could explore whether risk factors are similar in other populations and help better understand the nature of this relationship.

## Supporting information

**S1 Methods. Supplementary methods.**
(DOCX)

**S1 Table. Baseline characteristics of mothers and children enrolled into the disability compared to mothers and children not enrolled into disability study.**
(DOCX)

**S2 Table. Table reporting frequency of responses for each question of the Washington Group/UNICEF Module on Child Functioning (2015 version) from HIV unexposed infants.**
(DOCX)

**S3 Table. TREND document.**
(DOC)

**S1 Fig. Bar chart of functional difficulty prevalence by cut off in this study and field testing.**
(DOCX)

**S2 Fig. Study protocol.**
(DOCX)

**S3 Fig. PLOS1 global inclusivity questionnaire.**
(DOCX)

## Acknowledgments

Many thanks are offered to:

• All mothers and babies who participated in SHINE, without whom this work would not have been possible.

• Staff and Leadership of the Ministry of Health and Child Care in Chirumanzu and Shurugwi districts and Midlands Province for their input in operationalisation of study procedures.

• Ministry of Local Government officials in each district for facilitating field operations.

• The data and safety monitoring board: Professor Simon Cousens (Chair), Professor Hilda Mujuru, and Dr. Tariro Makadzange.

• The staff at Zvitambo including Mrs Virginia Sauramba for managing compliance issues and Mrs Phillipa Rambanepasi and her team for managing finances.

• To our program officers at the Bill & Melinda Gates Foundation and UK Aid for their dedication and input throughout the SHINE trial period.

## Author Contributions

**Conceptualization:** Andrew J. Prendergast, Jean H. Humphrey, Melissa J. Gladstone.

**Data curation:** Jaya Chandna, Florence Majo, Naume Tavengwa, Batsirai Mutasa, Bernard Chasekwa, Robert Ntozini, Andrew J. Prendergast, Jean H. Humphrey, Melissa J. Gladstone.

**Formal analysis:** Thomas Frederick Dunne, Jaya Chandna, Bernard Chasekwa, Robert Ntozini, Andrew J. Prendergast.

**Funding acquisition:** Andrew J. Prendergast, Jean H. Humphrey.

**Investigation:** Andrew J. Prendergast, Jean H. Humphrey.

**Methodology:** Thomas Frederick Dunne, Jaya Chandna, Florence Majo, Naume Tavengwa, Bernard Chasekwa, Robert Ntozini, Andrew J. Prendergast, Jean H. Humphrey, Melissa J. Gladstone.

**Project administration:** Thomas Frederick Dunne, Jaya Chandna, Florence Majo, Naume Tavengwa, Batsirai Mutasa, Andrew J. Prendergast, Jean H. Humphrey.

**Resources:** Jean H. Humphrey.

**Supervision:** Andrew J. Prendergast, Melissa J. Gladstone.

**Validation:** Jaya Chandna, Florence Majo, Bernard Chasekwa, Robert Ntozini, Melissa J. Gladstone.

**Writing – original draft:** Thomas Frederick Dunne.

**Writing – review & editing:** Thomas Frederick Dunne, Jaya Chandna, Florence Majo, Naume Tavengwa, Batsirai Mutasa, Bernard Chasekwa, Robert Ntozini, Andrew J. Prendergast, Jean H. Humphrey, Melissa J. Gladstone.

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
