## [Decision Letter · Decision Letter 0]

18 May 2022

PONE-D-21-31744Performance of the UNICEF/UN Washington Group tool for identifying functional difficulty in rural Zimbabwean childrenPLOS ONE

Dear Dr. Dunne,

Thank you for submitting your manuscript to PLOS ONE. After careful consideration, we feel that it has merit but does not fully meet PLOS ONE’s publication criteria as it currently stands. Therefore, we invite you to submit a revised version of the manuscript that addresses the points raised during the review process.

The reviewers had a number of minor concerns that must be addressed, including the need for clarification in the presentation of statistics and methodology. The reviewers' comments can be viewed in full, below and in the attached file.

We look forward to receiving your revised manuscript.

Kind regards,

Natasha McDonald, PhD

Associate Editor

PLOS ONE

Journal Requirements:

3. Please include a complete copy of PLOS’ questionnaire on inclusivity in global research in your revised manuscript. Our policy for research in this area aims to improve transparency in the reporting of research performed outside of researchers’ own country or community. The policy applies to researchers who have travelled to a different country to conduct research, research with Indigenous populations or their lands, and research on cultural artefacts. The questionnaire can also be requested at the journal’s discretion for any other submissions, even if these conditions are not met.  Please find more information on the policy and a link to download a blank copy of the questionnaire here: https://journals.plos.org/plosone/s/best-practices-in-research-reporting. Please upload a completed version of your questionnaire as Supporting Information when you resubmit your manuscript.

Reviewers' comments:

Reviewer's Responses to Questions

**Comments to the Author**

1. Is the manuscript technically sound, and do the data support the conclusions?

Reviewer #1: Yes

Reviewer #2: Partly

2. Has the statistical analysis been performed appropriately and rigorously? 

Reviewer #1: Yes

Reviewer #2: Yes

3. Have the authors made all data underlying the findings in their manuscript fully available?

Reviewer #1: Yes

Reviewer #2: Yes

4. Is the manuscript presented in an intelligible fashion and written in standard English?

Reviewer #1: Yes

Reviewer #2: Yes

5. Review Comments to the Author

Reviewer #1: The article is interesting and well-written. Moreover, the statistical analysis is well-conducted. I have only the following comment.

1. The statistical methods considered by the authors, with a special focus on the regression approaches, rely on assumptions about the nature of the underlying data. If the data do not meet those assumptions, then the results often are not valid. Therefore, it is important for authors to check if those assumptions are satisfied for the data at hand, at least, approximately.

Reviewer #2: Comments to the Authors

Abstract

• Abstract section needs some editorial corrections as indicated in the track changes

• The structure of the abstract should follow Plose one’s format

• Indicate the design of the study clearly as indicated in the body of the manuscript

• Mention briefly all statistical analyses used

Methods:

• Manuscript should stick to Plose one’s format

• Explain how pregnancy was confirmed

• “The ECD assessment took 2-3 hours and was conducted in the child’s home”. How the investigators overcome resistance that may arise from the respondent’s side?

• Nothing mentioned in the methods section how nutritional indices and dietary diversity were measured unless it was described in other papers. Explain

Results

• Are well organized.

• However, make clear whether the odds ratio mentioned in table 4 is the adjusted one or not.

• Lines 356-370 should appear before Table 5

Discussion

• “Disability was defined by the WGCFM, with concurrent validity assessed using the MDAT as a comparator”. Do we need definition of concept in discussion section?

• The authors should paraphrase the key messages in the first paragraph of the discussion as per the specific objectives of the study before commencing any discussion while highlighting on objectives of the study.

• Summarize discussion of each key finding in a single paragraph

• Implication of each key finding was not discussed in detailbeyond theoretical discussion

• The limitation of the study needs to be described in detail.

Conclusion and recommendations should be bold enough

6. PLOS authors have the option to publish the peer review history of their article (what does this mean?). If published, this will include your full peer review and any attached files.

Reviewer #1: No

Reviewer #2: **Yes: **Gudina Egata

---

## [Author Response · Author response to Decision Letter 0]

14 Jun 2022

Dear Natasha McDonald, PHD / PLOS ONE,

Many thanks for reviewing our article and for your comments and suggestions. Please see the uploaded document titled 'Response to Reviewers' for our reply and response to each of your / the reviewer's comments in turn. 

Kindest regards

Tom

Dr Thomas Dunne - corresponding author

---

## [Decision Letter · Decision Letter 1]

2 Sep 2022

Performance of the UNICEF/UN Washington Group tool for identifying functional difficulty in rural Zimbabwean children

PONE-D-21-31744R1

Dear Dr. Dunne,

We’re pleased to inform you that your manuscript has been judged scientifically suitable for publication and will be formally accepted for publication once it meets all outstanding technical requirements.

Kind regards,

Dr Joseph Donlan

Senior Editor

PLOS ONE

Additional Editor Comments (optional):

Reviewers' comments:

Reviewer's Responses to Questions

**Comments to the Author**

1. If the authors have adequately addressed your comments raised in a previous round of review and you feel that this manuscript is now acceptable for publication, you may indicate that here to bypass the “Comments to the Author” section, enter your conflict of interest statement in the “Confidential to Editor” section, and submit your "Accept" recommendation.

Reviewer #1: All comments have been addressed

2. Is the manuscript technically sound, and do the data support the conclusions?

Reviewer #1: Yes

3. Has the statistical analysis been performed appropriately and rigorously? 

Reviewer #1: Yes

4. Have the authors made all data underlying the findings in their manuscript fully available?

Reviewer #1: Yes

5. Is the manuscript presented in an intelligible fashion and written in standard English?

Reviewer #1: Yes

6. Review Comments to the Author

Reviewer #1: (No Response)

7. PLOS authors have the option to publish the peer review history of their article (what does this mean?). If published, this will include your full peer review and any attached files.

Reviewer #1: No

---

## [Editor Report · Acceptance letter]

7 Sep 2022

PONE-D-21-31744R1 

Performance of the UNICEF/UN Washington Group tool for identifying functional difficulty in rural Zimbabwean children 

Dear Dr. Dunne:

I'm pleased to inform you that your manuscript has been deemed suitable for publication in PLOS ONE. Congratulations! Your manuscript is now with our production department. 

Kind regards, 

on behalf of

Dr Joseph Donlan 

Staff Editor

PLOS ONE